

# In leukemia, knock-down of the death inducer-obliterator gene would inhibit the proliferation of endothelial cells by inhibiting the expression of *CDK6* and *CCND1*

Honghua Cao[1,*], Lilan Wang[1,*], Chengkui Geng[2], Man Yang[3], Wenwen Mao[4], Linlin Yang[5], Yin Ma[1], Ming He[1], Yeying Zhou[1], Lianqing Liu[1], Xuejiao Hu[1], Jingxing Yu[6], Xiufen Shen[7], Xuezhong Gu[8], Liefen Yin[6] and Zhenglei Shen[1]

[1] Department of Hematology, The Third Affiliated Hospital of Kunming Medical University, Kunming, China
[2] Department of Orthopedics, Yan'an Hospital of Kunming City, The Affiliated Hospital of Kunming Medical University, Kunming, China
[3] Department of Endocrinology, The Affiliated Hospital of Yunnan University & The Second People's Hospital of Yunnan Province, Kunming Yunnan, Kunming, Yunnan, China
[4] Department of Geriatics, The Second Hospital of Kunming, Kunming, China
[5] Department of Gynecology, The Third Affiliated Hospital of Kunming Medical University, Kunming, China
[6] Department of Hematology, The Second Affiliated Hospital of Kunming Medical University, Kunming, China
[7] Department of Laboratory, The Second Affiliated Hospital of Kunming Medical University, Kunming, China
[8] Department of Hematology, The First People Hospital in Yunnan Province, Kunming, China
* These authors contributed equally to this work.

Corresponding authors
Liefen Yin, ylfynkm@126.com
Zhenglei Shen, szl1020@163.com

## ABSTRACT

**Background:** Endothelial cells (ECs) are a critical component of the hematopoietic niche, and the cross-talk between ECs and leukemia was reported recently. This study aimed to determine the genes involved in the proliferation inhibition of endothelial cells in leukemia.

**Methods:** Human umbilical vein endothelial cells (HUVEC) were cultured alone or co-cultured with K562 cell lines. GeneChip assays were performed to identify the differentially expressed genes. The Celigo, MTT assay, and flow cytometric analysis were used to determine the effect of RNAi DIDO on cell growth and apoptosis. The differently expressed genes were verified by qRT-PCR (quantitative real-time PCR) and western-blot.

**Results:** In K562-HUVEC co-cultured cell lines, 323 down-regulated probes were identified and the extracellular signal-regulated kinase 5 (ERK5) signaling pathway was significantly inhibited. Among the down-regulated genes, the *death inducer-obliterator gene* (*DIDO*) is a part of the centrosome protein and may be involved in cell mitosis. As shown in the public data, leukemia patients with lower expression of *DIDO* showed a better overall survival (OS). The HUVEC cells were infected with *shDIDO* lentivirus, and reduced expression, inhibited proliferation, and increased apoptosis was observed in *shDIDO* cells. In addition, the expression of

*Cyclin-Dependent Kinase 6* (*CDK6*) and *Cyclin D1* (*CCND1*) genes was inhibited in *shDIDO* cells. Finally, the public ChIP-seq data were used to analyze the regulators that bind with *DIDO*, and the H3K4me3 and PolII (RNA polymerase II) signals were found near the Exon1 and exon2 sites of *DIDO*.

**Conclusion:** The knock-down of *DIDO* will inhibit the proliferation of endothelial cells in the leukemia environment. The expression of *DIDO* may be regulated by H3K4me3 and the inhibition of *DIDO* may lead to the down-regulation of *CDK6* and *CCND1*. However, how *DIDO* interacts with *CDK6* and *CCND1* requires further study.

## INTRODUCTION

The increased number of circulating endothelial cells (ECs) in the peripheral blood was detected in multiple myeloma (*Zhang et al., 2005*), myelodysplasia (*Cortelezzi et al., 2005*), and acute myeloid leukemia patients (*Wierzbowska et al., 2005*). Besides, the higher level of circulating ECs and endothelial precursor cells (EPCs) was associated with more aggressive disease and shorter survival (*Rigolin et al., 2010*) in chronic lymphocytic leukemia.

The cross-talk between leukemia cells and endothelial cells was reported recently. The endothelial cells (ECs) provide a fertile niche that will promote the proliferation of primitive and aggressive leukemia cells (*Gunsilius et al., 2000*). Besides, ECs could elaborate on angiocrine factors, which will take part in the reconstitution of normal and malignant stem/progenitor cells (*Butler, Kobayashi & Rafii, 2010*). ECs provide critical support for the survival and progression of leukemia stem cells (LSCs) (*Le et al., 2021*), which would promote regeneration of leukemia (*Kaplan, Rafii & Lyden, 2006*; *Colmone et al., 2008*). On the other hand, the leukemic blasts could secrete numerous cytokines, which will augment the proliferation of microvascular endothelial cells in primary acute myelocytic leukemia (AML) cells (*Hatfield et al., 2012*). Leukemia may induce the activation of resting ECs and these activated ECs would protect the leukemia cells from chemotherapy injury (*Pezeshkian et al., 2013*). Due to the protective effect of ECs in leukemia microenvironment (*Bosse et al., 2016*), vascular targeted drugs may be a new strategy for AML treatment decisions.

Leukemia-derived ECs may originate from bone marrow-derived hemangioma blast progenitor cells (*Bobryshev, Orekhov & Chistiakov, 2015*), as the BCR-ABL fusion transcript in ECs was found derived from bone marrow progenitor cells (*Gunsilius et al., 2000*). However, the AML cells could integrate into vasculature and fuse with ECs in *vivo*, and the AML cells could differentiate into endothelial-like cells *in vitro* (*Cogle et al., 2014*).

A better understanding of the interaction between ECs and leukemia may inspire the design of innovative therapies for leukemia. Niche target treatment may help restore damaged vascular microenvironment, increase chemotherapy delivery and increase treatment responses. To determine the targets which will inhibit the ECs may be a new direction. In this study, we tried to investigate the genes involved in the interaction
between the ECs and leukemia cells by GeneChip. The proliferation of HUVEC cells was inhibited when co-cultured with the K562 cells. A *death inducer obliterator gene* (*DIDO*) showed lower transcript abundance in HUVEC-K562 co-cultured cells. Although we have not found a report about the mechanism of *DIDO* in leukemia, there were reports showing that *DIDO* was involved in the development of solid tumors, such as bladder cancer (*Li et al., 2020*), RCC (*Xiao et al., 2020*), and melanoma (*Braig & Bosserhoff, 2013*). In this study, we investigated the role of *DIDO* in endothelial cells in the leukemia environment.

## MATERIALS AND METHODS

### Cell lines and cell culture

Human umbilical vein endothelial cells (HUVEC) and human myeloid leukemia cell line (K562) were purchased from the American Type Culture Collection (Rockville, MD). The cell lines were maintained using RPMI (Roswell Park Memorial Institute, Buffalo, NY, USA) 1,640 medium (Gibco Co, Waltham, MA, USA) supplemented with 10% FBS at 37 °C in a humidified atmosphere containing 5% $CO_2$. The HUVEC and K562 cell lines were mixed and co-cultured for 4 days. CCK-8 Cell Counting Kit-8 (CCK-8) was used to determine the cell viability by colorimetric assays at 450 nm (*Tang et al., 2019*).

For RNA extraction and analysis of cell proliferation and apoptosis, the suspension of K562 cells was removed. Then the HUVEC cells in the two groups (HUVEC *vs.* HUVEC-K562) were washed by PBS buffer and collected for further experiments. The HUVEC cells were digested with pancreatin before RNA extraction.

### Plasmid constructs and transfection

For gene knockdown, the GV115 vector was used in this study, which used the green fluorescent protein as a reporter gene, and the multiple cloning sites were driven by a human U6 promoter. The *DIDO* was targeted by the shRNA sequences of 5′-GGATGAGACTCATTCAGAA- 3′. The sequence was cloned into the multiple cloning sites by restriction enzyme of AgeI and EcoRI. Plasmid transfection was performed as a former study (*Nabzdyk et al., 2011*). The cell lines were seeded into 96-well plates. After transfection for 2~3 d, the GFP was observed under a fluorescence microscope. The cells were used for further studies when the cell density in the wells reached 70–90%.

### Celigo and MTT assays

Cells were inoculated into the 96-well plates (2,000 cells/well) and three repeats were taken. The cell numbers were measured by Celigo Imaging Cytometer (*Nabzdyk et al., 2011*) and the numbers were recorded for 5 days. The cell numbers were normalized to the cell numbers on the first day after seeding.

MTT (3-(4,5-dimethylthiazol-2-yl)-2,5-diphenyl tetrazolium bromide) assay (*Moodley et al., 2014*) was performed to analyze the proliferation of cells. A 1 mg/mL MTT was added to each well and incubated at 37 °C for 4 h. Then the culture medium was removed and the DMSO (150 μl) was added into each well, and then the plate was shaken for 3 min.

The Tecan Infinite M2009PR plate reader was used to measure the absorbance at 490 nm/570 nm.

## Cell apoptosis analysis

The cell apoptosis was measured following the manufacturer's instructions of Annexin-FITC Apoptosis Detection Kit (BD Biosciences, Franklin Lake, NJ, USA). Cells were cultured in a 96-well plate for 3–5 days in the 37 °C incubator, and then the cells were harvested and washed in PBS. Cells were added to 0.5 ml binding buffer and Annexin V-FITC, then the cells were stained in the dark for 15 min at room temperature. Cells stained by Annexin V-FITC were considered apoptotic cells (*Pan et al., 2014*) which were measured by a BD Accuri™ C6 flow cytometer (BD Biosciences, Franklin Lakes, NJ, USA).

To analysis the cell apoptosis, Caspase 3/7 enzyme activity was measured by Caspase-Glo® 3/7 Assay (Promega, G8091, Madison, WI, USA). Caspase-Glo 3/7 reagent was added to the sample with a volume ratio of 1:1, and the cells were incubated for another 1 h at 37 °C. The Tecan Infinite M2009PR plate reader was used to detect the luminescence in each well at 490 nm/520 nm (*Stennicke & Salvesen, 1997*).

## Angiogenesis analysis

The serum-free supernatants of tumor cells from different experimental groups were collected and suspended the HUVEC cells to $2 \times 10^4$ cells/100 uL. After being cultivated at 37 °C for 4–6 h, the angiogenesis assay was performed by the Celigo instrument.

## Microarray processing and data analysis

The samples were hybridized with the GeneChip microarrays (901838; Affymetrix, Santa Clara, CA, USA) to determine gene expression abundance according to the manufacturer's instructions. The expression profile was preprocessed by the Limma package in Bioconductor.

A robust multiarray averaging algorithm was used to perform background correction, quantile normalization, and probe summarization on the microarray data to obtain a gene expression matrix. The cut-off for the background correction was 20%, and the coefficient of variation was 25%. The Benjamini-Hochberg method was used to correct the significant difference level (FDR). The screening criteria for significantly different genes were: |Fold Change| > 1.5 and FDR < 0.05 (*Ritchie et al., 2015*). The biological pathways analysis of genes was performed by Ingenuity Pathway Analysis (IPA).

## RNA extraction and qRT-PCR analysis

According to the manufacturer's protocol, the Trizol reagent (Invitrogen, Waltham, MA, USA) was used to extract total RNA from frozen cells. For cDNA synthesis, 1 µg of total RNA was used to synthase the cDNA by the Go Script reverse transcription system (Promega, Maddison, MA, USA). The genes were detected by the SYBR Master Mixture (DRR041B; Takara, Kusatsu, Shiga, Japan) using the LightCycler480 Real-Time PCR system (Roche, Basel, Switzerland). For qRT-PCR, the GAPDH gene was used as endogenous control. The primers sequences and the length of the amplifications were

shown in Table S1. The $2^{-\Delta\Delta Ct}$ method was used to calculate the fold change for gene expression relative to the control.

## Protein extraction and Western-blot analysis

Total protein was isolated from cells using protein cell lysis buffer and extracted by centrifugation at 13000 rpm for 20 min at 4 °C. The equal amount of whole cell lysate was separated by SDS-PAGE gel electrophoresis. After the proteins were transferred to the PVDF membranes (Bio-Rad, Hercules, CA, USA), the membranes were blocked by 5% skimmed milk and immunoblotted with the primary antibodies at 4 °C. Then the membranes were blotted with the secondary antibodies at room temperature for 1 h. The following primary antibodies were used: anti-DIDO (1:1000, HPA049904, Sigma), anti-CCND1 (1:500, Cat2978, CST), anti-CDK6 (1:500, Cat3136, CST), anti-GAPDH (1:2000, Sc-32233, Santa Curz). The secondary antibodies were anti-rabbit or anti-mouse IgG conjugated to horseradish peroxidase (Santa Cruz Biotechnology, Dallas, TX, USA). The Dyne ECL STAR Western Blot Detection kit (Dyne Bio, Seoul, Korea) and a chemiluminescent image system (Fusion Solo system, Villber Lourmat, France) were used to analyze the protein abundance.

## Statistical analysis

The data were shown as the mean ± S.D. from three independent replicates. The student's t-test was performed to analyze the quantitative data. $P < 0.05$ was considered statistically significant.

# RESULTS

## GeneChip microarrays analysis of HUVEC and K562-HUVEC co-cultured cell lines

In this study, the human umbilical vein endothelial cells (HUVEC) were used as endothelial cells models *in vitro*. When the HUVEC cells were co-cultured with the K562 leukemia cell lines for 4 days, the proliferation was inhibited significantly (Fig. S1). Then we analyzed the gene expression changes in HUVEC cells when co-cultured with the K562 leukemia cell lines by GeneChip, to investigate the genes which will inhibit the endothelial cells' proliferation in leukemia progression.

Compared with HUVEC lines, 398 probes up-regulated expression and 323 probes down-regulated expression in K562 co-cultured HUVEC lines (Fig. S2A). Ingenuity Pathway Analysis (IPA) found that the extracellular signal-regulated kinase 5 (ERK5) signaling was significantly inhibited (Z-score = −2.111) (Fig. S2B). On the other hand, the DEGs were mainly enriched in microtubule dynamics (Z-score = 2.783), migration of brain cancer cell lines (Z-score = 2.549), liver tumor (Z-score = −2.782) and cell death of mononuclear leukocytes (Z-score = −2.561) (Fig. S2C).

## Construction of RNAi cell lines and cell proliferation analysis

We selected the first 30 down-regulated expression genes (log2 (change fold) > 1, $P < 0.05$) for further analysis (Table S2). RNAi lentiviral vectors for these 30 genes were constructed
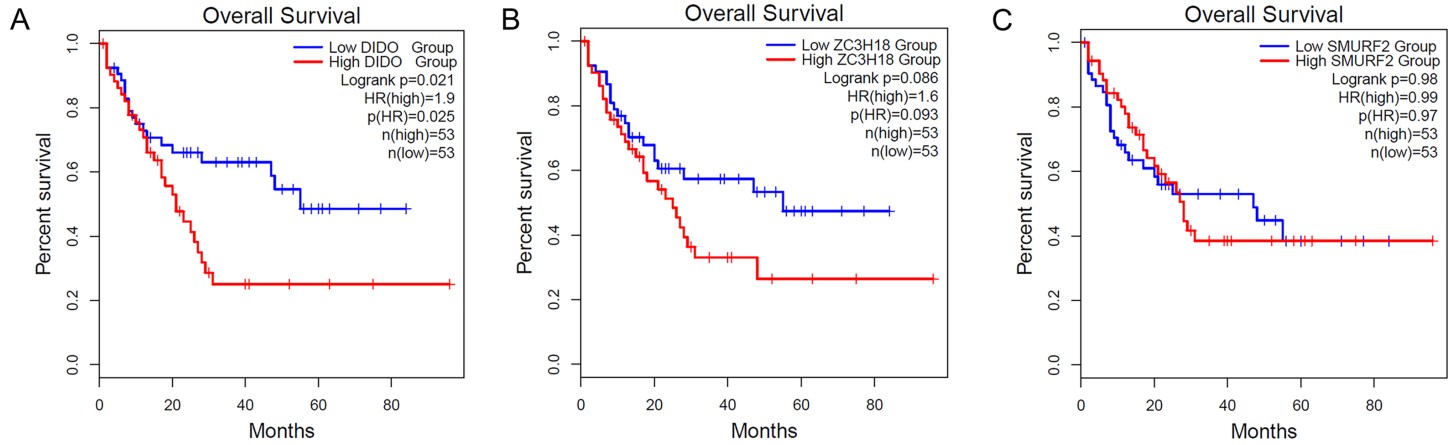

**Figure 1** The KM-plot in Acute myeloid leukemia stratified by the expression level of DIDO (A), ZC3H18 (B), and SMURF2 (C).

and transfected into HUVEC cells. A total of 22 transgenic cell lines were successfully obtained, including the negative control (NC) and positive control (PC). The cell count results showed that the proliferation of cells was normal in the NC group, and which was significantly inhibited in the PC group. The proliferation folds on the fifth day were 12.09 and 2.12 times higher than those on the first day in the NC and PC groups, respectively. The fold change (FC) of cell count ([FC in NC group on the 5th day compared to which on the 1st day]/[FC in experiment group on the 5th day compared to which on the 1st day]) was used to evaluate the influence of gene RNAi in cell proliferation. The proliferation of *shDIDO*, *shZC3H18*, and *shSMURF2* cell lines was significantly inhibited, and the change fold was 3.37, 2.54, 2.07, respectively (Fig. S3).

## Patients with lower transcript abundance of DIDO showed a better overall survival

To determine the effect of the *DIDO*, *ZC3H18*, and *SMURF2* in leukemia patients, we analyzed the survival based on their expression status from the public data (http://gepia2.cancer-pku.cn/). As shown in Fig. 1A, the acute myeloid leukemia (AML) patients with the lower *DIDO* expression level, showed a better overall survival (HR = 1.9; *P* = 0.025). However, the different expressions of *ZC3H18*, and *SMURF2* did not affect the overall survival in AML patients (Figs. 1B and 1C).

## The proliferation of *shDIDO* cell line is inhibited and the apoptosis is increased

To further investigate the function of the *DIDO* (Death inducer obliterator) gene in endothelial cells, we analyzed the proliferation and apoptosis of *shDIDO* cells. Firstly, the expression of *DIDO* in *shDIDO* cells was analyzed. qRT-PCR found that the expression level of the *DIDO* gene at the mRNA level was suppressed in *shDIDO* cell lines (*P* < 0.05), and the reduction efficiency reached 95.1% (Fig. 2A). Western-blot detection found that the expression of DIDO protein in the *shDIDO* cells decreased by four times, compared with the *shCtrl* group (Fig. 2B).

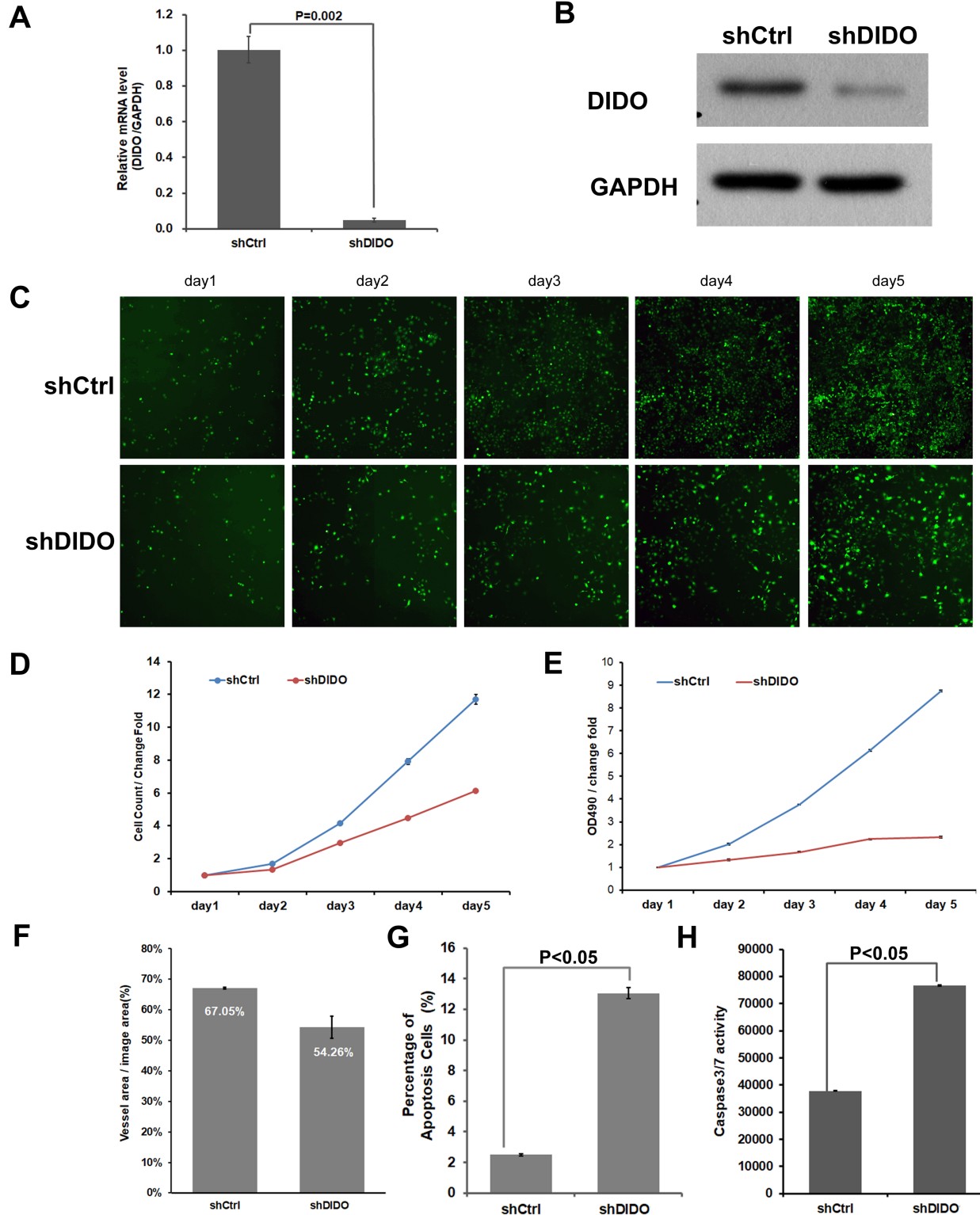

**Figure 2 The function assays of *DIDO* gene.** (A) The RNA transcript abundance of *DIDO* gene in *shDIDO* and *shCtrl* cells detected by qRT-PCR; (B) Western-blot assay for the expression of DIDO protein; (C) Cell proliferation pictures of *shDIDO* and *shCtrl* cell lines by Celigo; (D) and (E) Cell count of *shDIDO* and *shCtrl* cell lines; (F) Analysis of the area of blood vessel formation; (G) Cell apoptosis ratio analyzed by FACS; (H) Caspase3/7 activity assays.

The proliferation rate of shDIDO cell line was analyzed by Celigo (Figs. 2C and 2D) and MTT (Fig. 2E), and the proliferation rate of the *shDIDO* cells was significantly decreased. This may indicate that the *DIDO* gene is significantly related to the proliferation ability of HUVEC cells.

The number of cells in the apoptotic state was detected by Annexin V-APC single staining method, and it was found that apoptosis cells in *shDIDO* group increased significantly than the HUVEC cells ($P < 0.05$) after 5 days (Fig. 2G). Additionally, by detecting the activity of caspase, it was found that the activity of caspase3/7 in the *shDIDO* group was significantly increased. These results indicate that the *DIDO* gene was significantly related to the apoptosis of HUVEC cells (Fig. 2H).

Due to the importance of angiogenesis in tumor progression, we analyzed the effect of *DIDO* gene depletion on angiogenesis. The ability of *shDIDO* cells to form lumens was analyzed to investigate the metastasis ability of tumors. It was found that the area of angiogenesis-related blood vessels in the *shDIDO* group were 20% less than that in the shCtrl group ($P < 0.05$) (Fig. 2F), which indicates that the *DIDO* gene may not associated with HUVEC cells angiogenesis.

## GeneChip analysis of shDIDO and shCtrl cell lines

To investigate the biological pathways *DIDO* involved, the GeneChip expression profiles of *shCtrl* and *shDIDO* cell lines were analyzed. It was found that 521 genes in the *shDIDO* cells were up-regulated and 1,006 genes were down-regulated (Fold Change > 1.5 and FDR < 0.05), compared with *shCtrl* cells (Fig. 3A, Table S3). IPA analysis found that the ERK/MAPK signaling was significantly inhibited (Z-score = −2.041) (Fig. 3B). Additionally, the functions including morbidity or mortality (Z-score = 6.734) and organismal death (Z-score = 6.709), were significantly activated. The functions including cell viability (Z-score = −5.369), cell survival (Z-score = −5.349) were significantly suppressed (Fig. 3C and Table S4).

The genes down-regulated in *shDIDO* cells, and which are involved in tumorigenesis and development, were selected for further analysis (Table S5). Among them, 30 probes were identified by qRT-PCR as their expression pattern were similar to the GeneChip (Table S6).

DIDO is a part of the centrosome protein and plays an important role in spindle assembly, so we infer that the *DIDO* gene may correlate with the cell cycle. We further analyzed the cell cycle regulation genes of *Cyclin Dependent Kinase 6* (*CDK6*) and *Cyclin D1* (*CCND1*). The transcription of *CCND1* and *CDK6* in the *shDIDO* cells were 0.364 and 0.404 times of the *shCtrl* group, respectively. Meanwhile, the western-blot analysis found that the protein expression levels of CCND1 and CDK6 in *shDIDO* cells were reduced by 81.3% and 58.1%, respectively. However, the DIDO maybe not interact with CDK6 or CCND1 directly, as shown in Fig. 3D. We searched the protein interaction database and did not find the direct interaction of DIDO with *CDK6* or *CCND1* either. The genes that may directly interact with *DIDO* were *SRSF1*, *SRPK2*, *EED*, and *WWP2* in the interaction network of *DIDO* analyzed by IPA.

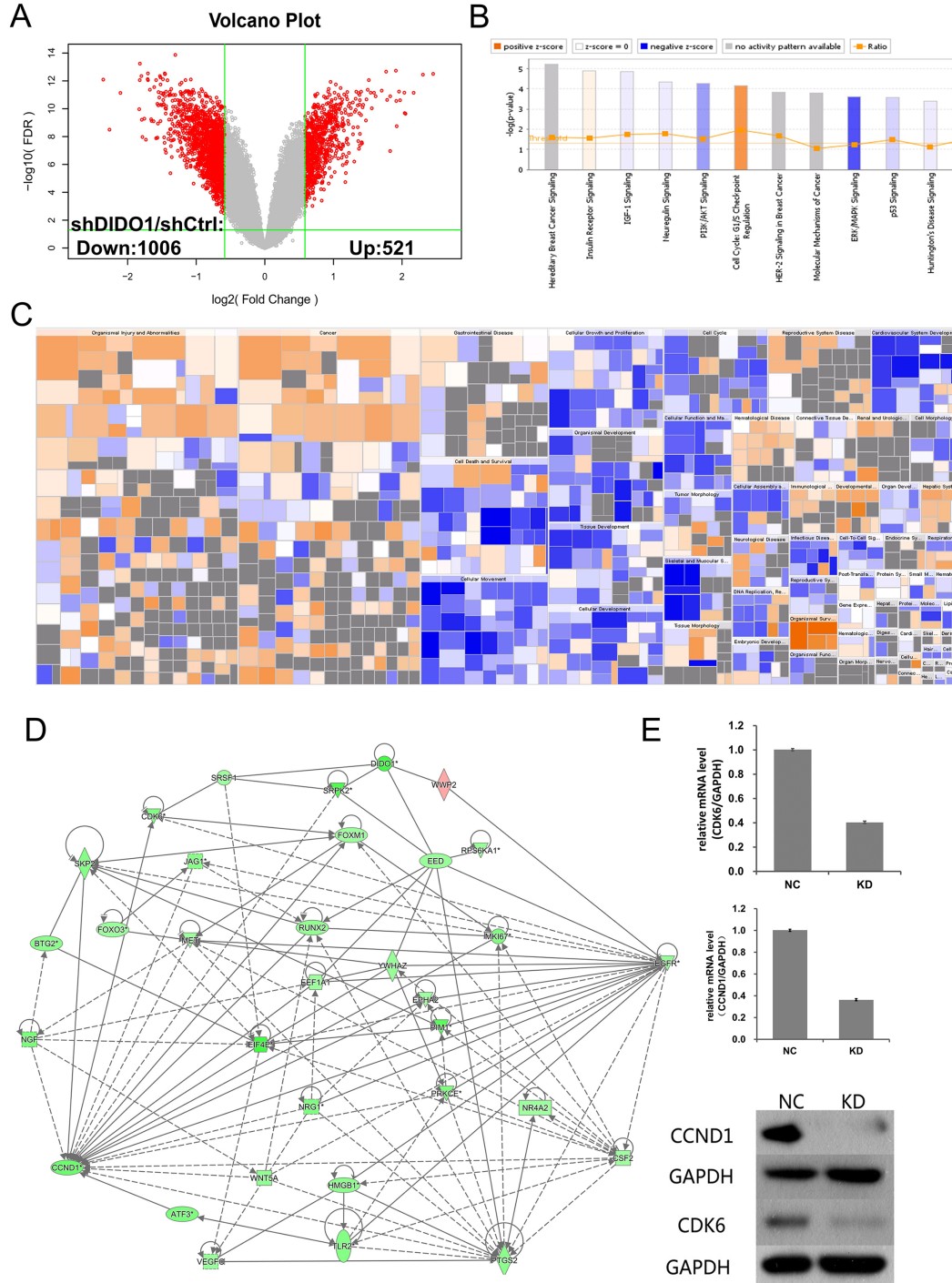

**Figure 3** **Analysis of downstream genes regulated by *DIDO*.** (A) The volcano map of the differently expressed probes in *shCtrl* and *shDIDO* cell lines; (B) The enrichment of DEGs in the classical signal pathway; (C) Disease and function heat maps show the expression changes of DEGs in different diseases and functions. Orange means the disease or functional state is activated (Z-score > 0), blue means the disease or functional state is inhibited (Z-score < 0), and gray means the disease or functional state is not determined (Z-score cannot be calculated); The disease or function is significantly activated if Z-score > 2, and significantly inhibited if Z-score < −2. Significantly-activated diseases or functions include: morbidity or mortality (Z-score = 6.734), organic death (6.709), etc.; Significantly-inhibited diseases or functions include: cell viability (−5.369), cell survival (−5.349); (D) Gene interaction network diagram shows the interaction network between molecules; (E) The expression of *CCND1* and *CDK6* analyzed by qRT-PCR and Western-blot.

## DISCUSSION

Endothelial cells provide a fertile niche that allows for the propagation of primitive and aggressive leukemic clones. This study aimed to identify the genes involved in the interaction between endothelial cells and leukemia.

Firstly, the GeneChip assay showed that the expression of 323 probes was down-regulated in K562-HUVEC co-cultured cells (Fig. S3A), and the ERK5 signaling was significantly inhibited. It has been reported that the ERK5 pathway mediates cell survival, apoptosis, and proliferation signaling in embryonic stem cells (*Williams et al., 2016*). We infer that the decreased ERK5 signaling may correlate with the proliferation inhibition of HUVEC cells. There is increasing evidence to indicate that the ERK5 signaling takes part in the development and progression of several types of cancers, including breast cancer, myeloma, lymphoma, leukemia (*Stecca & Rovida, 2019*). In addition, some studies suggested that the ERK5 could represent a promising target for therapeutic intervention in leukemia (*Kang et al., 2018*).

To investigate the key genes involved in the inhibition of HUVEC cells when co-cultured with K562, RNAi cell lines of the top 30 down-regulated expression genes were constructed and we analyzed the proliferation of them. The proliferation of *shDIDO*, *shZC3H18*, and *shSMURF2* cells was significantly inhibited, and their transcripts were significantly inhibited in HUVEC-K562 co-cultured cells. These may indicate that the down-regulate expression of these genes would help inhibit the activity of ECs in the leukemia environment. To understand the role of these genes *in vivo*, we analyzed the survival of leukemia patients when stratified by the expression abundance. It was found that leukemia patients with lower expression of *DIDO* showed better survival. Based on these findings, we focused this study on the *DIDO* gene.

DIDO plays an important role in mitotic progression and chromosome instability as it is a component of the centrosome proteins and plays an essential role in spindle assembly (*Xiao et al., 2020*). It has been reported that *DIDO* is related to chromosomal instability. *DIDO* gene may be a novel MSI biomarker, as its mutation has a high concordance level with MSI-H status (microsatellite instability high), based on research enrolled 1,301 colorectal cancer FFPE (formalin-fixed, paraffin-embedded) tissue sections (*Velasco et al., 2021*). There are increasing evidence showing that the *DIDO* plays an important role in tumor onset and progression. In bladder cancer, the reduction of *DIDO* mRNA resulted in increased apoptosis, reduced proliferation *in vitro*, and inhibited tumorigenesis *in vivo*. The authors pointed that the potential mechanism of *DIDO* action might involve SAPK/JNK signaling cascades (*Li et al., 2020*). In addition, in melanoma cells, *DIDO* was found to induce the expression of integrin α, and promoting the attachment, migration, invasion and apoptosis resistance of melanoma cells (*Braig & Bosserhoff, 2013*). In this study, the transcript and protein abundance of *DIDO* gene was inhibited in siRNA cell lines, which resulted in an inhibition of proliferation, and an up regulation of apoptosis. This is consistent with the bladder cancer and melanoma.

To investigate the genes that are affected by *DIDO*, the DEGs between *shDIDO* and *shCtrl* cell lines were analyzed. The ERK/MAPK signaling was significantly inhibited in

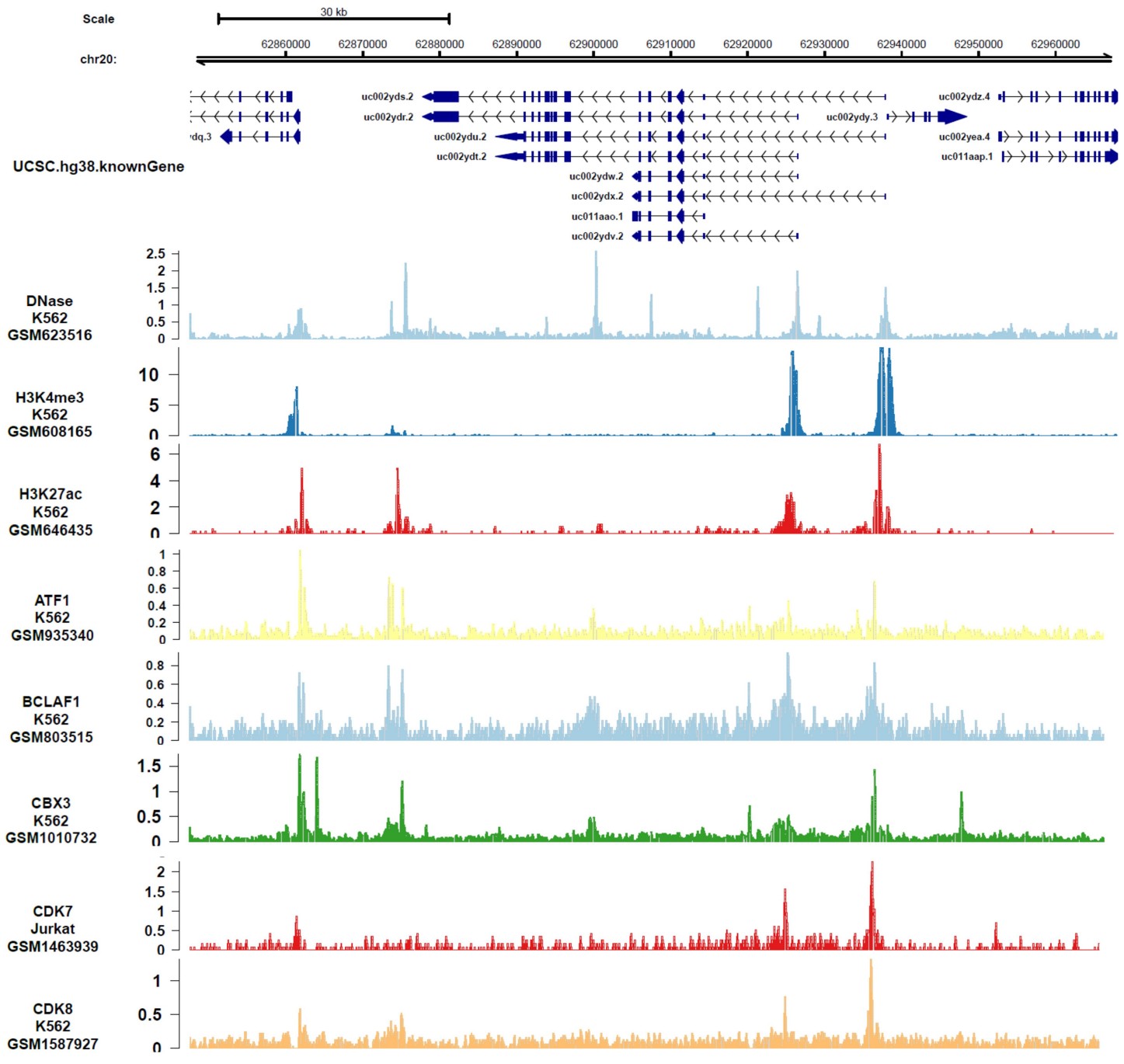

**Figure 4** The transcript factors and epigenetic factors that bind with *DIDO* gene analyzed by public data.

both the HUVEC-K562 co-cultured cells and *shDIDO* cells, the ERK5 pathway mediates apoptosis and proliferation signaling in several kinds of tumor cells (*Stecca & Rovida, 2019*). It was reported that the ERK5 was regulated by phosphorylation and established a link between the CDK pathway during mitosis (*Iñesta-Vaquera et al., 2010*). On the other hand, the *cyclin D1* gene is a key step in cell proliferation, and it may be a novel target of the

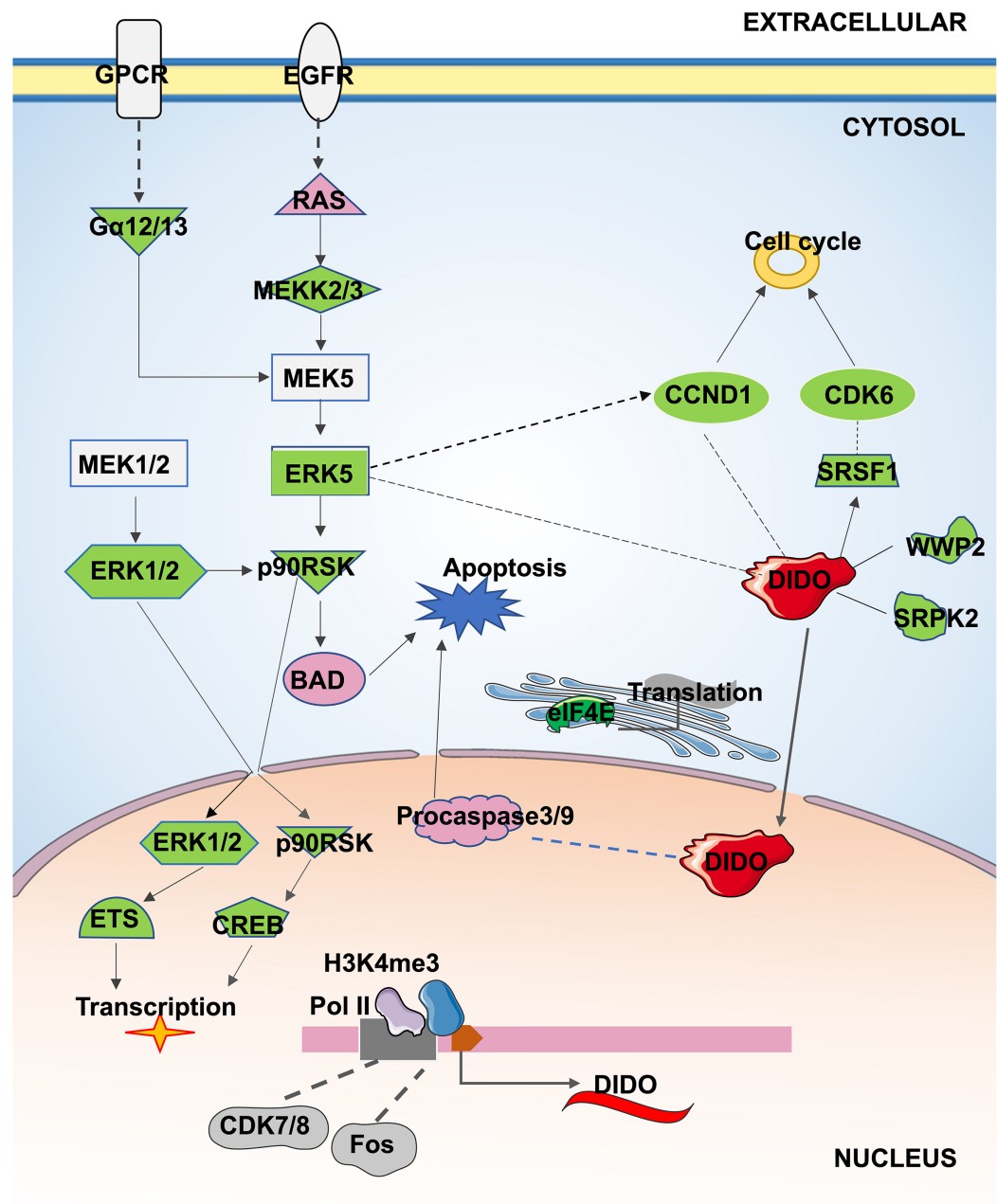

**Figure 5 The apoptosis and proliferation mechanism in leukemia endothelial cells that may be regulated by *DIDO*.** Green represents down-regulated expression of genes in *shDIDO* cells, and pink represents up-regulated expression of genes in *shDIDO* cells. The solid line indicates a clarified relationship between genes; The dotted line indicates the relationship between genes requiring further study.

ERK5 cascade (*Mulloy et al., 2003*). In this study, we also found that the expression of cell cycle genes, *CDK6* and *CCND1*, were down-regulated in *shDIDO*, which was verified by qRT-PCR and western-blot. According to these results, there may be crosstalk between *DIDO*, *ERK5*, *CDK6*, and *CCND1*, and these genes may work together to inhibit cell proliferation. However, how these genes interact with each other still need further study.

As *DIDO* may act as a transcription factor (*Rojas et al., 2005*), we screened for potential *DIDO* target genes that down regulated in *shDIDO* cells. The IPA interaction network indicated that *DIDO* directly interact with *WWP2*, *SRPK2* and *SRSF1*. *WWP2* (WW domain containing E3 ubiquitin protein ligase 2) gene encodes a protein that play a role in the regulation of oncogene signaling pathways *via* interactions with SMAD proteins and the tumor suppressor PTEN. *WWP2* could promote the proliferation of gastric cancer cells in a PTEN-dependent manner, and its silencing will inhibit proliferation and growth of gastric cancer cells (*Wang et al., 2020*), suggesting a vital role of *WWP2* in cancer progression. *SRPK2* (Serine/Arginine-Rich Protein-Specific Kinase-2, SRSF protein kinase-2) is up-regulated in multiple human tumors, and plays an important role in the progression and metastasis of prostate cancer (*Zhuo et al., 2018*). *SRSF1* (serine/arginine-rich splicing factor 1) promotes proliferation and injury-induced neointima formation in vascular smooth muscle cells (*Xie et al., 2017*), and it could promote tumorigenesis through regulation of alternative splicing in colon cancer (*Chen et al., 2017*), glioblastoma (*Zhou et al., 2019*), and other cancers. The overexpression of *SRSF1* could promote cell proliferation and delay cell apoptosis during acinar morphogenesis in breast cancer (*Anczuków et al., 2012*). According to these previously researches, *DIDO* may play roles by interaction with *WWP2*, *SRPK2* and *SRSF1*.

In order to analyze the genes or epigenetic modification that may regulate the expression of *DIDO*, the published ChIP-seq data in different leukemia cell lines (including K562) was analyzed. There are multiple DNase-seq peaks near *DIDO*, presuming that there are regulatory factors binding at the corresponding position. As shown in Fig. 4 and Fig. S4, the *CDK7*, *CDK8*, *ATF1*, *BCLAF1*, and *CBX3* had binding peaks at the transcription start site (TSS) near exon 1 and exon 2. H3K4me3 and Pol II (POLR2A) bind to the site near *exon1* and *exon2* of *DIDO* gene. The *EGR1*, *FOS*, *MAX*, *NCOR1*, H3K4me1, H3K27ac, *MED1*, and *EP300* had binding peaks specifically at the TSS on exon1, and *CHD7*, *SIRT6*, *c-MYC* indicate binding signals near the exon 2 TSS. This indicate that the transcript of *DIDO* gene was regulated by the transcript factors and epigenetic factors. However, the correlation between these factors and the *DIDO* gene needs further experimental verification.

## CONCLUSIONS

In conclusion, the apoptosis and proliferation mechanism that may be regulated by *DIDO* in leukemia endothelial cells was summarized in Fig. 5. The ERK5 signal will be inhibited by the down-regulation of *DIDO* in *shDIDO* cell lines, and the genes in ERK5 signaling may play roles in cell apoptosis and proliferation, and regulate the gene transcription and translation. In addition, the inhibited *DIDO* in HUVEC will indirectly inhibit the expression of *CDK6* and *CCND1*, which will inhibit the proliferation of cells.

The inhibition of the proliferation of endothelial cells may inhibit the development of leukemia, and inducing cell apoptosis may become a therapy to treat leukemia. The *DIDO* gene discovered in this study provides a theoretical basis for the development of drug targets for leukemia. However, it is also necessary to study the gene expression of *DIDO* in leukemia patients.

### Funding

This study was supported by the National Natural Science Foundation of China (NO. 81360089), the Applied Basic Research in Yunnan Province of China (NO. 2019FE001-067, NO. 2017FE468-204), the Project of Educational Commission of Yunnan Province of China (NO. 2018JS229), the Applied Basic Research in Yunnan Province of China (NO. 202001AY070001-070), and the High-level Talents Training support program in Yunnan Province (YNWR-MY-2020-015). The funders had no role in study design, data collection and analysis, decision to publish, or preparation of the manuscript.

### Grant Disclosures

The following grant information was disclosed by the authors:
National Natural Science Foundation of China: 81360089.
Applied Basic Research in Yunnan Province of China: 2019FE001-067, 2017FE468-204.
Project of Educational Commission of Yunnan Province of China: 2018JS229.
Applied Basic Research in Yunnan Province of China: 202001AY070001-070.
High-level Talents Training Support Program in Yunnan Province: YNWR-MY-2020-015.

### Competing Interests

The authors declare that they have no competing interests.

### Author Contributions

- Honghua Cao conceived and designed the experiments, performed the experiments, prepared figures and/or tables, authored or reviewed drafts of the paper, and approved the final draft.
- Lilan Wang conceived and designed the experiments, performed the experiments, prepared figures and/or tables, authored or reviewed drafts of the paper, and approved the final draft.
- Chengkui Geng conceived and designed the experiments, performed the experiments, prepared figures and/or tables, and approved the final draft.
- Man Yang analyzed the data, authored or reviewed drafts of the paper, and approved the final draft.
- Wenwen Mao analyzed the data, prepared figures and/or tables, and approved the final draft.
- Linlin Yang analyzed the data, prepared figures and/or tables, and approved the final draft.
- Yin Ma analyzed the data, prepared figures and/or tables, and approved the final draft.
- Ming He analyzed the data, prepared figures and/or tables, and approved the final draft.
- Yeying Zhou analyzed the data, prepared figures and/or tables, and approved the final draft.

- Lianqing Liu analyzed the data, authored or reviewed drafts of the paper, and approved the final draft.
- Xuejiao Hu analyzed the data, authored or reviewed drafts of the paper, and approved the final draft.
- Jingxing Yu performed the experiments, authored or reviewed drafts of the paper, and approved the final draft.
- Xiufen Shen performed the experiments, authored or reviewed drafts of the paper, and approved the final draft.
- Xuezhong Gu performed the experiments, authored or reviewed drafts of the paper, and approved the final draft.
- Liefen Yin conceived and designed the experiments, authored or reviewed drafts of the paper, and approved the final draft.
- Zhenglei Shen conceived and designed the experiments, authored or reviewed drafts of the paper, and approved the final draft.

## Data Availability

Data is available at NCBI GEO, accession number: GSE156713.

## Supplemental Information

Supplemental information for this article can be found online at http://dx.doi.org/10.7717/peerj.12832#supplemental-information.

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
