# Peer review of "In leukemia, knock-down of the death inducer-obliterator gene would inhibit the proliferation of endothelial cells by inhibiting the expression of CDK6 and CCND1"

_PeerJ, doi:10.7717/peerj.12832_

## Round 0.1 · original submission · Major Revisions

Dear Dr. Shen,
Our reviewers have suggested some major corrections and have requested additional data and modifications in your current manuscript. Before your manuscript can be considered for publication please address each of those concerns.

·

Basic reporting

The manuscript “The knock-down of Death inducer-obliterator1 gene (DIDO1) would promote apoptosis and inhibit proliferation of leukemia induced Endothelial cells by inhibiting the expression of CDK6 and CCND1” by Cao et al. describes the effect of inhibition of DIDO1 on the proliferation of Leukemia induces endothelial cells. GeneChip analysis was performed to find out gene upregulation and downregulation. The effect of DIDO1 gene on cell proliferation and apoptosis has been analyzed. The study is interesting and can be useful in increasing survival of leukemia patients.

The manuscript needs significant improvement. It has to be improved for both English language and grammar as well as organization. It should be thorough and easily understandable. Even the results in the Abstract section loses its flow and does not provide the complete picture of the manuscript. The abbreviations should be introduced when they are used first time.

Experimental design

Is there any in vivo data showing that the knockdown of DIDO1 gene would either treat leukemia or increase survival? In vivo data will add a great value to the study.

Did the authors analyze different domains in DIDO1? For example, if the effect is same when NLS deleted DIDO1 is used. This will elaborate the mechanism of action of DIDO1.

Validity of the findings

Figures 1C and S2 are not clear. It seems that GFP is higher in shDIDO cells. It should mean that the cells proliferation is higher in these cells as compared to control. If it is so, it will contradict the entire results.

Did authors use an activator of CDK6 or CCND1 gene to check if there is any counter effect on DIDO1 action? It will confirm the mechanism of action of DIDO1.

All the GeneChip data can be seen in supplementary. Authors have written that 30 genes were analyzed by qPCR for the expression level determination. That data should be included.

Authors have discussed the inhibitory effect of DIDO1 on ERK/MAPK signaling pathway, however except the GeneChip analysis there is no other evidence provided. It will be great if authors can provide more qPCR/protein expression/phosphorylation data in control and shDIDO cells. Also, more can be discussed about this pathway how it can be beneficial in leukemia patients.

In schematic figure 4, it is not clear how or at which step DIDO1 is affecting ERK/MAPK signaling pathway. It shows general pathway, how does it relate to the current study?

Reviewer 2 ·

Basic reporting

The manuscript needs substantial English language editing.

The background information for the molecule of interest, Dido1, is not sufficiently described.

Figures legends need to be more descriptive and axes need better definition
Ex: Fig 1G, Percentage can be revised to Percentage cell death

Raw data are shared.

Experimental design

The experimental system leading to the most critical data with Genechip is not described. It is not clear how the co-cultured cells were prepared for GeneChip analysis.

Validity of the findings

The importance of the molecule of interest Dido1 in the context of leaukemia-activated endothelial cells is unclear. The authors are suggested to discuss the potential role of Dido in endothelial cells.

Additional comments

In the manuscript by Cao et.al. the authors compare gene expression profiles of endothelial cells and leukemia-activated endothelial cells. They identified Dido gene expression to be downregulated in leukemia-educated HUVEC cells. They independently confirmed this using stable silencing of Dido1 in HUVEC cells and found that it resulted in cell death. They used publicly available data to identify that Dido may regulate cell cycle genes CCND1 and CDK6.
Listed below are the major concerns regarding this manuscript in its current form:
1. The authors have not detailed the method for co-culture of HUVEC cells with K562. It is recommended to give a detailed description of the procedure used prior to gene expression profiling by Genechip.
Were the HUVEC cells separated from K562 for the GeneChip experiment?
2. The authors do not describe clearly the cell type in which the genes were upregulated or down regulated.
3. The authors do not sufficiently introduce the background information for Dido necessary to convince the audience of the importance of this protein in leukemia or endothelial biology. Previous results indicate that Dido is required for cancer cell proliferation (Neoplasma Vol.67, No.5, p.1074–1084, 2020) The authors are suggested to explain the relevance of Dido in the endothelial cells. The implications of this finding for cancer biology should be discussed.
Further, Dido1 has been described as a tumor suppressor gene. How does this current data align with previous literature on Dido1?
4. With the knockdown of Dido 1 with shRNA, do the other splice variants (Dido2, Dido3) get affected?
5. It is recommended that the authors revise the manuscript to check for
- typographical errors: such as Genichip (Ln 41),
- expand acronyms: such as MVD (Ln 82)
- usage of biological jargons: such as leukemic induced ECs.(Ln 40)
- sentences are fragmented and do not flow well. For instance: “In this study, we aimed to identify the genes function in the formation of leukemic induced ECs.”(Ln 39-40)
This current phrasing of sentences makes it hard to follow. It is suggested that the authors use professional English language editing services to improve the manuscript.

---

## Round 0.2 · Major Revisions

As the authors have suggested there are still more concerns that need to be addressed.

·

Basic reporting

The revised submission of the manuscript entitled “The knock-down of Death inducer-obliterator gene would promote apoptosis and inhibit proliferation of leukemia induced Endothelial cells by inhibiting the expression of CDK6 and CCND1”, concerning the effect of DIDO1 gene knockdown on leukemia induced Endothelial cells. Authors have answered all the comments/concerns point-wise and improved the manuscript accordingly.

English language must be improved further. At some places DIDO1 has mentioned, while at others DIDO. It should be consistent throughout the manuscript including figures.

Experimental design

no comments

Validity of the findings

Authors have adjusted contrast in figure 2C. However, contrast and other parameters should remain identical in control vs the experimental pictures. Is it possible to take new pictures if the authors have saved the slides?

Reviewer 2 ·

Basic reporting

The sentence construction still needs to be improved. The statements and ambiguious and sometime conflicting.

Additional reference are need to clarify basic concepts, please see additional comments.

The replacement of DIDO1 to DIDO has to be consistent in all of the Figures and legends. This change was made only to the text, as suggested in the review report.

Experimental design

Methods are not clearly described. Please see additional comments

Validity of the findings

No comment.

Additional comments

The authors have submitted an updated version of the manuscript addressing some of the points raised in the review report. The origin of cell lines has been described in the manuscript.
Listed below are some of the concerns that still remain:

1. The authors use various terminologies regarding leukemia activated endothelial cells which make for a difficult read. At times the statements are even conflicting.
The authors use “leukemia induced endothelial cells” which they need to clarify. Do they refer to leukemia induced endothelial cell activation or differentiation?
In the title the authors mention “DIDO knockdown inhibits proliferation of leukemia induced endothelial cells”.
In the abstract the mention “formation of endothelial cells to leukemia” (Ln40)
Later in the text, the authors state “mechanism of how leukemia cell induces ECs formation”

The literature recognizes the cross talk between endothelial cells and leukemic cells and details two separate phenomenon:
1. Leukemia mediated endothelial cell activation : Ref https://www.ncbi.nlm.nih.gov/pmc/articles/PMC3613371/
2. Endothelial to hematopoietic transition
https://pubmed.ncbi.nlm.nih.gov/27015586/

The authors need to clarify what they refer to in the above listed statements with suitable references.

3. The authors described the cell used for Genechip, as requested. But it still lacks key details. The authors reported that the GeneChip analysis was performed with HUVEC cells vs HUVEC-K562 co-cultured cells. Did the authors wash away the K562 suspension cells before harvesting cells from the co-culture or use the entire co-culture set up for gene expression analysis ? Please specify.

4. Similarly, for the proliferation assay reported in Figure S1, the authors would need to clarify the method used to ensure that only HUVEC cells were counted by Celigo and/or analyzed by MTT assay without the interference of K562 cells.

5. The shRNA targeting DIDO gene may target all three isoforms so the authors agreed to change this to DIDO rather than DIDO1, but the data and legend in all of the Figures still reports DIDO1. Please correct this throughout the manuscript.


6. In the final results section, the authors explore potential transcriptional regulators of the DIDO gene. However, the outstanding question from the manuscript is the mechanism by which Leukemic cell interaction may suppress the expression of DIDO in HUVEC cells. What do the authors speculate regarding this reported finding? It is suggested to include this in the discussion section.


Minor:
1. Check abbreviations :
CEC should be changed to EC in Ln 70,72
LAML should be changed to AML in Ln 213, 215
2. Check the statement “Human umbilical vein endothelial cells (HUVEC) are capable to differentiate into endothelial cells in vitro.” (Ln 181-182)

---

## Round 0.3 · Minor Revisions

Please address the comments by Reviewer 2.

Reviewer 2 ·

Basic reporting

No comment. Please see additional comments

Experimental design

No comment. Please see additional comments

Validity of the findings

No comment. Please see additional comments

Additional comments

In the revised manuscript, “In leukemia, knock-down of the death inducer-obliterator gene would inhibit the proliferation of endothelial cells by inhibiting the expression of CDK6 and CCND1”, the authors have addressed some of the concerns raised in the previous review report. While some changes in the rebuttal were satisfactory some other concerns remain. They acknowledged that some terminology in the manuscript was misleading to the readers and needed to be clarified. It appears they have corrected some of these instances however, the revised, track changed manuscript still contains significant instances of inaccuracies.
Listed below are concerns that still remain:
1. The authors were asked to clarify the phrase “ Leukemia-induced endothelial cells”. They have addressed this in the rebuttal, however this statement still remains in the revised manuscript in the conclusion section of the abstract Ln 56-57 “The knock-down of DIDO will inhibit the formation of leukemia-induced endothelial cells”.
Another instance in Ln 92-93 In this study, we investigated the role of DIDO in the formation of leukemia-induced ECs.”

2. There are instances of abbreviations that are not expanded OS for overall survival in the abstract Ln 50.

3. Among the changes that the authors have made, they added this statement to the discussion section “In this study, we found that there are H3K4me3 and Pol II (POLR2A) signals near Exon1 and exon2, suggesting that there are isoforms with Exon1 and exon2 as transcription start sites.”
This statement is factually incorrect about the DIDO gene alternate transcripts generated by splicing. It is clearly defined in the literature that DIDO2 and DIDO3 isoforms generated with the inclusion of exons 6-8 and 15-16 respectively near the 3’ end of the gene, rather than exon 1 and 2 as transcription start sites like the authors suggest. (Ref: https://www.jci.org/articles/view/24177 , Dido gene expression alterations are implicated in the induction of hematological myeloid neoplasms, by Futterer et al in 2005)

---

## Round 0.4 · accepted · Accept

Dear Dr. Shen, your revised manuscript has now been accepted. Congratulations.